# Fostering Project-Based Learning through Industry Engagement in Capstone Design Projects

Ashraf Badir [1] , Robert O'Neill [1], Kristoph-Dietrich Kinzli [2,*], Simeon Komisar [1] and Jong-Yeop Kim [1]

1 Department of Environmental and Civil Engineering, Florida Gulf Coast University, Fort Myers, FL 33965, USA
2 Department of Civil and Environmental Engineering, Colorado School of Mines, 1500 Illinois St., Golden, CO 80401, USA
* Correspondence: kkinzli@mines.edu

**Abstract:** Extensive studies have indicated that real-life project-based learning through industry involvement in capstone design courses provides benefits to students, faculty, and industry practitioners. This paper presents the contributions of industry participants to student experiential and project-based learning in the civil and environmental engineering senior design courses at Florida Gulf Coast University (FGCU). Surveys were conducted to obtain insights into the contributions of industry involvement in the capstone design course from the perspectives of both students and practitioners. Practitioners have been involved in various roles, including project mentors for capstone design projects and/or judges for students' capstone design project presentations. Practitioners, through the students, are provided with new ways of looking at and solving problems. Practitioners, through their involvement, provide valuable feedback to the faculty and students that enriches the overall experience gained in the capstone design course. This feedback helps improve student performance on their projects and provides them with additional tools to carry forward into their engineering careers. However, there was a gap in perception between students and practitioners with regard to the benefits of industry involvement. This paper also describes two successful capstone design projects and culminates success drivers from the reflection of instructors teaching these courses. The results of this study have substantial implications for faculty teaching these courses. They showed what students did well and pinpointed areas for improvement through the lens of industry practitioners.

**Keywords:** capstone; senior design; industry participation; project-based learning; practitioner involvement; experiential learning



## 1. Introduction

The profession of engineering takes the knowledge of mathematics and natural sciences gained through study, experience, and practice and applies this knowledge with judgment to develop ways to utilize the materials and forces of nature for the benefit of all humans [1]. The engineer applies his or her knowledge to design and develop usable devices, processes, and structures. The Accreditation Board of Engineering and Technology (ABET) formal definition of engineering design states that it is "*the process of devising a system, component, or process to meet desired needs. It is a decision-making process (often iterative), in which the basic sciences, mathematics, and the engineering sciences are applied to convert resources optimally to meet these stated needs.*" [2].

One important aspect of design education is the capstone design experience [3] featuring a real-life engineering project. Achieving a successful capstone design course in civil and environmental engineering is a critical task. It is generally challenging to replicate projects as encountered in design offices with the risk that capstone design courses become analysis courses [4]. Site conditions and local regulations frequently control civil and environmental engineering design. Faculty without day-to-day experience will not have the

same educational impact as practitioners working in the field. Consequently, a significant number of institutions engage industrial clients to sponsor capstone projects [5,6].

Industry involvement is a critical component of student learning due to the experience and knowledge obtained from the related activities [5,7]. In engineering education, although there is a high degree of agreement on the importance of professional skills (interpersonal skills, teamwork, communication, and problem-solving skills), employers have observed a big gap between expectation and reality [8,9]. This was one of the primary reasons for completing this study as one of our main goals as educators is to prepare engineering students for successful careers in industry. With regard to engineering practice, the literature has shown significant differences in the rigor of design and professionalism between school and work [10]. Therefore, an affiliation of active engineers and professionals from industry would strengthen the professional elements of engineering education [11].

The most common form of industry involvement in capstone courses may be through industry sponsorship of design projects, which often includes mentorship and funding of projects and project teams [12]. According to a 2005 study of capstone design courses in the United States, 71% of the courses included industry-sponsored projects [13]. Capstone faculty view the course as a means to provide students with an opportunity to apply what they have learned throughout their undergraduate career through an open-ended design project in an environment that simulates the real world [14]. The benefit of the involvement of practicing engineering professionals has been actively discussed in the literature [15–20]. One study [20] stated that constructible projects were possible when a strong partnership between the university and the municipality was in place. The authors argued that a constructed project offered excellent public relations opportunities for both the university and municipality, and built a cadre of new professionals who understood the complexity and bureaucracy of civil engineering projects. The students' interactions with professionals also developed a degree of professionalism in the students that was not generally present in graduating seniors [20].

Brunell [16] reported that the industry sponsors of the senior design program felt the time and money were well spent educating the engineers of the future, some of whom became their employees. They observed that the optimum group size of students was four, with one of those being proficient in drawing. Teams of less than four tended to be overwhelmed by the many components they were responsible to consider during the design process. Teams with more than four tended to have members who did not contribute sufficiently and the team tended to overwhelm the consultant [16].

Industry benefits by receiving technical assistance from senior students in performing preliminary analyses and designs to screen different candidate solutions and generate new ideas for solving existing problems [15]. The students benefit from working on real-world problems, interacting with professional engineers, and exposure to economical, legal, and regulatory constraints that are not discussed in conventional undergraduate courses with the possibility of being employed by their mentors after graduation [12,19]. Relevant, industry-partnered design is an important part of the undergraduate education experience for tomorrow's engineers [21]. Consequently, students gain valuable insights into the "nuts and bolts" of design in their field and acquire the skills needed to enter the practice [15].

Nevertheless, most of the studies on capstone design courses have mainly concentrated on structure, pedagogy, assessment, and course outcomes [10]. Limited studies have addressed the perspectives of both students and practitioners with respect to the benefits of industry engagement and possible differences between the two groups [22]. The studies that do exist have addressed student perception and success and industry involvement [12,23,24]. This paper aims at filling this gap in the literature by identifying the contributions of industry participants to student experiential learning from the perspectives of both students and practitioners. FGCU is a teaching-focused university located in a mid-size city, with 80% of the classes taught by full-time faculty.

## 2. Contextual Background of the Senior Capstone Courses

This paper describes both civil and environmental engineering capstone design courses mentored by faculty and practitioners. The capstone engineering design course at the Department of Environmental and Civil Engineering at FGCU is taught in three sections during the spring semesters; two civil engineering sections and one environmental engineering section. The average total course enrollment in the four spring semesters (2014–2017) was 56 students. The course met once a week for a period of 2 h and 45 min. The focus of the course is on a real-life design project with usually 3–4 students per team, with about 16 projects per academic year. Students formed their own teams, a strategy that minimizes possible conflicts throughout the semester. Practitioners mentored more than half of the projects, students proposed some projects, and faculty assigned others. The practitioners were local civil and environmental engineering professionals with at least ten years of experience. Most of the practitioners were licensed (e.g., professional engineers) or appropriately certified (e.g., American Institute of Architects (AIA), American Planning Association's Certified Planner (AICP)). The role of the practitioners was to mentor the students through the design process including project initiation, the scope of work, project planning, preliminary, intermediate, and final design work as well as final presentations.

Tables 1 and 2 show representative project topics adopted with collaboration from industry in civil engineering and environmental majors, respectively. Various projects were performed by students in both civil engineering and environmental engineering majors in the past years. Faculty typically seek projects from the networks of local practitioners in the fall semester and had a list of projects with associated practitioner mentors ready at the first week of the capstone design classes in the spring semester. Students then formed their own teams and selected a project to work on throughout the spring semester. The capstone projects were real-world design-oriented projects that were in their conceptual stage. As such, in addition to data provided by the mentor or mentor's organization, students may have to survey and collect data from project sites and various sources. Most of these projects were not financially sponsored by the practitioner mentors. Practitioners played a major role in assisting and mentoring students throughout the semester. In addition, they evaluated/graded the final poster presentations of the student teams as judges at the end of the semester.

**Table 1.** Representative civil engineering projects sponsored by practitioners.

| Topic | Scope |
| --- | --- |
| A space network control center | Design an airport communication building and antenna foundation |
| Design of an access road | Traffic pattern research, design of pavement, road geometry, retention ponds, control structures, and cost analysis |
| A university athletic field | Design of site (basketball courts, football, and soccer fields) and an underground water detention system |
| A two-story school building | Design a steel structure and its foundation |
| A water control structure | Water management research, design of adjustable weir to facilitate both water storage and flood prevention |
| Culverts for everglades restoration | Water management research, design roadway culverts to induce the spreading of sheet flow |
| Public park | Design of open space, amphitheater, drainage, access, and event area |

**Table 2.** Representative environmental engineering projects sponsored by practitioners.

| Topic | Scope |
|---|---|
| Design of a phosphorus recovery process for wastewater and stormwater treatment | Design and test electrocoagulation system using solar power and aluminum electrodes |
| Design of a bioreactor to treat campus-generated hazardous wastes | Design and build a hybrid suspended and attached growth system |
| Evaluation of management strategies to lower energy usage in a reverse osmosis desalination plant | Reconfigure the number and placement of membranes to utilize pressure recovery |
| Examination of a wetland system nutrient removal in an urban setting | Design and place monitoring wells and weirs to measure the quality and quantity of water |
| Source tracking of microbial contaminants in an urban stream | Design a sampling and analysis protocol to track fecal contamination |
| Design of a floating wetland for control of nutrients in a stormwater detention pond | Specify the size of floating islands, number and type of plants |

Student performance was assessed through both individual work effort (20%) and team work effort (80%). The major individual effort was the individual design portfolio. The team work effort consisted of in-class interim design oral presentations (20%), interim and final written reports (30%), and final design poster presentations (30%). The final design poster presentations were judged by industry practitioners based on a rubric provided by faculty. With the rubric, practitioners evaluated both verbal presentation skills (organization, delivery, and professionalism) and written presentation skills (content and quality of poster) of student teams. The accreditation of engineering programs necessitates design experience and recognizes the value of industry partners in the capstone design courses in preparing students for careers in engineering fields [5]. Table 3 presents the course learning objectives and the corresponding ABET student outcomes. The ABET outcomes in the paper are the old a-k outcomes as this study was conducted before the change to the new 1–7 outcomes [2]. These course objectives were assessed by the project deliverables submitted and/or presented as individual and teamwork efforts mentioned above.

**Table 3.** Course learning objectives.

| Course Objectives | ABET Student Outcomes |
|---|---|
| Applying appropriate mathematical and scientific models to solve client-based problems | a |
| Designing a system, component, or processes to meet desired engineering needs | c and k |
| Determining the impact of contemporary issues on the design process considering realistic constraints such as economic, environmental, social, political, ethical, health and safety, regulatory, manufacturability, and sustainability | j |
| Developing an understanding and being able to explain the importance of professional and ethical responsibility, and professional development | f |
| Demonstrating effective communication skills | g |
| Demonstrating an understanding of how their solutions impact global, social, and environmental contexts | h |

## 3. Feedback from Students and Practitioners

Feedback from both students and practitioners was collected to gain some insights into the industry involvement in these capstone courses. A survey questionnaire was designed and sent to all 26 practitioners who were mentors, judges, and/or clients between 2014 and 2017 and encompassed four capstone cohorts. This practitioner survey was designed to understand the purpose and benefits of their involvement, the ability of students in capstone design, open-ended questions about their mentoring experience, and suggestions for improvement. For the last year of the study period, questions regarding the benefits

of industry involvement were also added to the official end-of-semester student survey administered by the university, called student perception of instruction (SPoI).

Figures 1 and 2 display the role of practitioners and the number of capstone design projects that they had mentored. Most of the practitioners were involved as mentors (7%) or both mentors and judges (40%) (Figure 1). In addition, 57% of all of the practitioners involved mentored three projects or more. This indicated that they were involved in these capstone design courses for several years and some practitioners mentored more than one project at the same time.

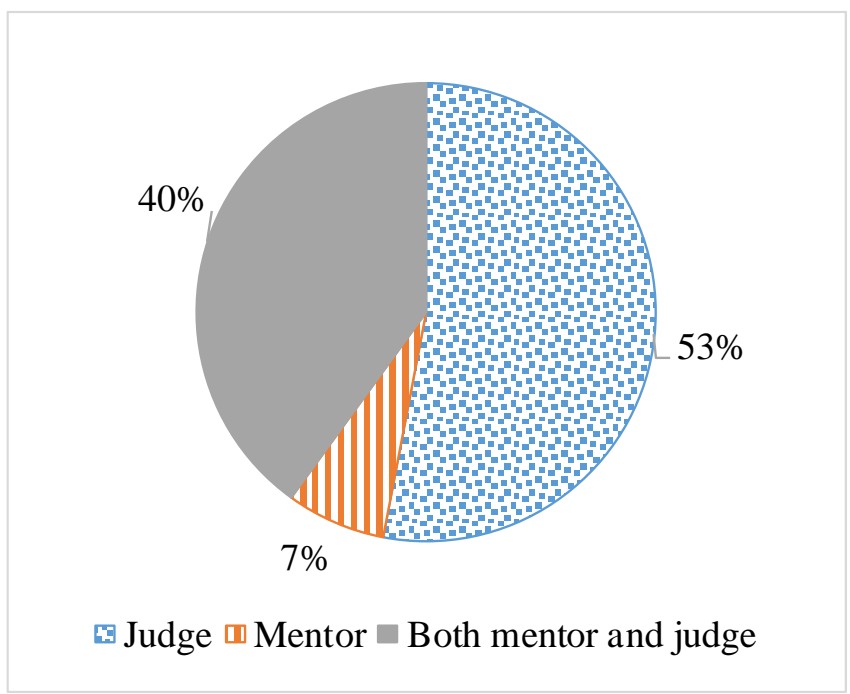

**Figure 1.** Roles of practitioners.

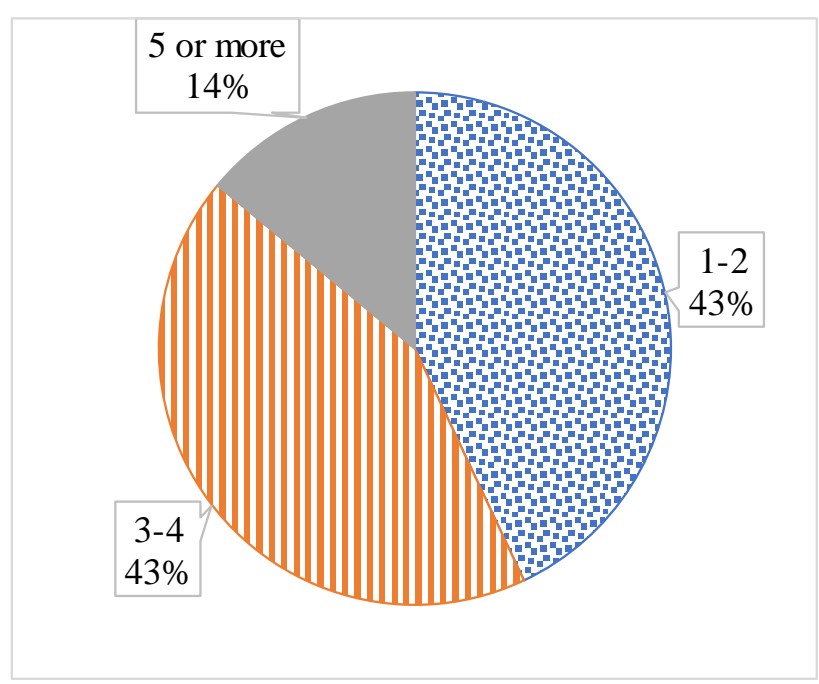

**Figure 2.** Number of projects mentored.

Although the motivation for professionals to get involved varied, all of them agreed or strongly agreed that they participated in the training of new engineers (Figure 3). Less than half of practitioners agreed or strongly agreed with other purposes such as "*gaining access to a pool of graduating engineers for recruitment*" (40%) and "*advertising practitioner's organization on campus*" (33%) (Figure 3).

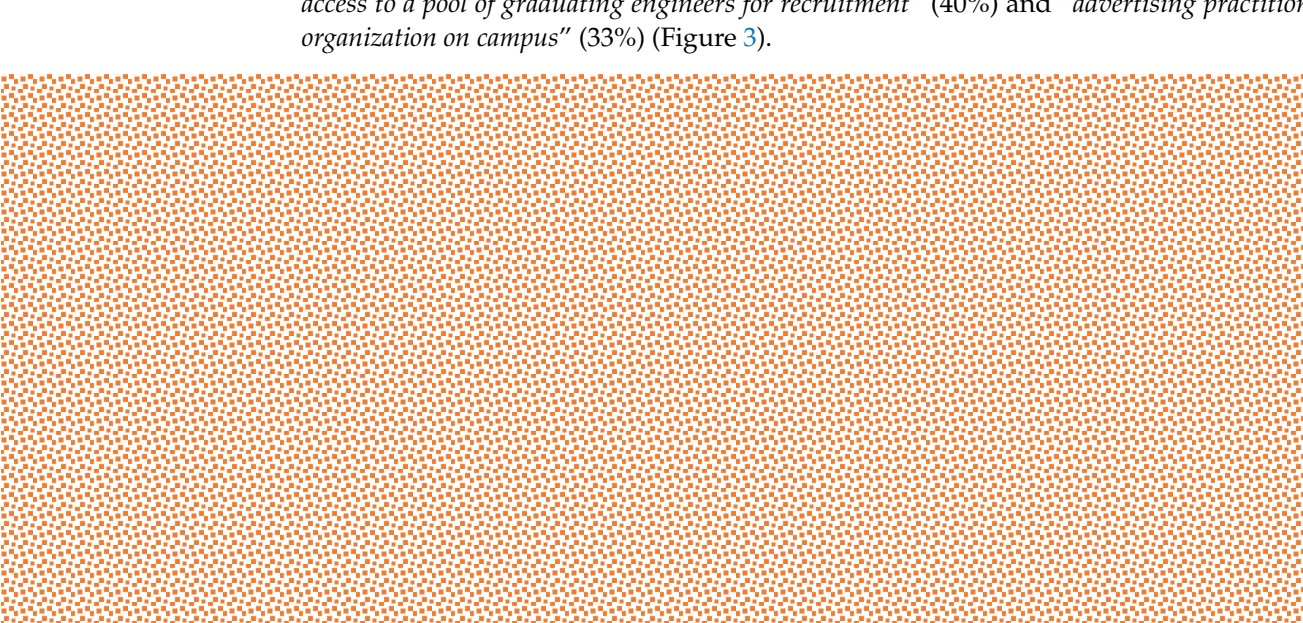

**Figure 3.** Purpose of industry practitioners' involvement.

Figure 4 illustrates the students' and practitioners' perceptions of the benefits to practitioners and their organizations. The level of agreement seemed different between students and practitioners in the statement "*mentor or mentor's company receives additional technical resources dedicated towards solving a technical problem at a lower cost*" (Figure 4a). While 62% of students either agreed or strongly agreed with the statement, only 40% of practitioner respondents agreed or strongly agreed with this same statement. However, similar proportions of practitioners (67%) and students (69%) either agreed or strongly agreed that working with engineering students can provide practitioners' organizations with a new way of looking at and solving problems (Figure 4b).

The perceptions of students and practitioners as to the benefits to students due to industry involvement were somewhat different (Figure 5). Two major benefits to students due to industry involvement that almost all practitioners agreed or strongly agreed with were: (i) bridging the gap between what is learned in the engineering curriculum and what is expected of graduates when they work in the industry (94%, Figure 5a); (ii) exposing students to professional practice (100%, Figure 5b). Smaller proportions of students either agreed or strongly agreed with these benefits, i.e., 77% for the first and 69% for the second benefit (Figure 5a,b). In contrast, while more than two-thirds (69%) of students agreed or strongly agreed with the benefit that "*students acquire or practice many of the professional skills that are used on a daily basis by design engineers and practitioners,*" only half (53%) of practitioners agreed or strongly agreed with that (Figure 5c). This implied that other experiential learning such as internships may better realize the latter benefit.

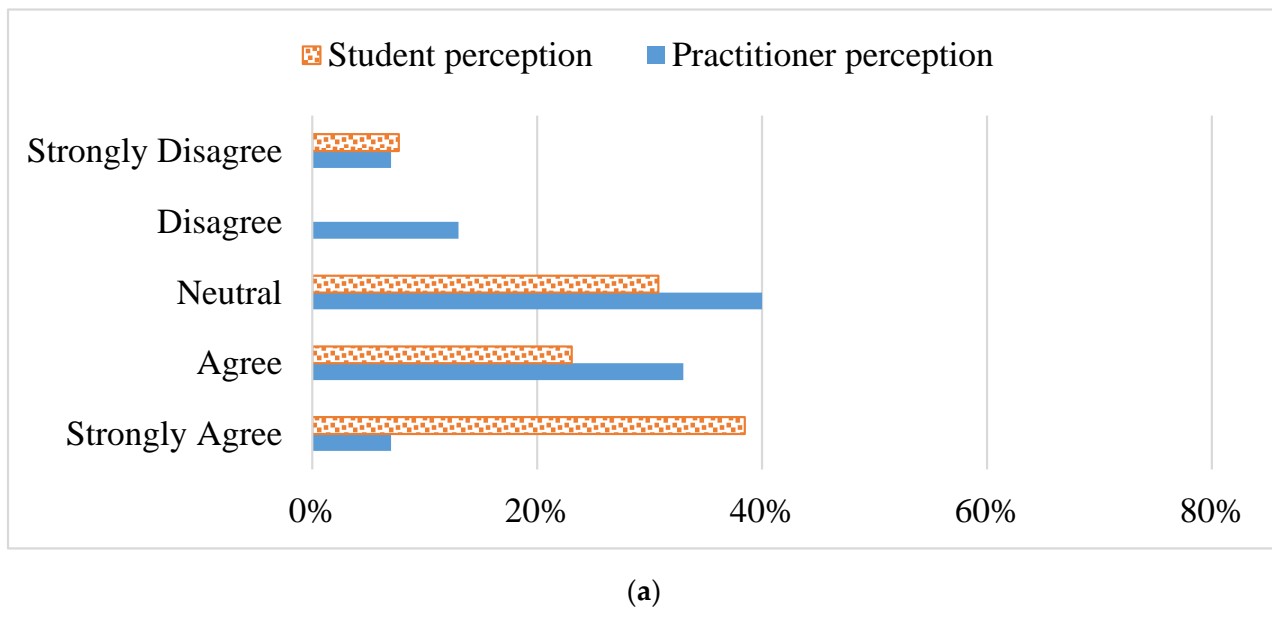

(**a**)

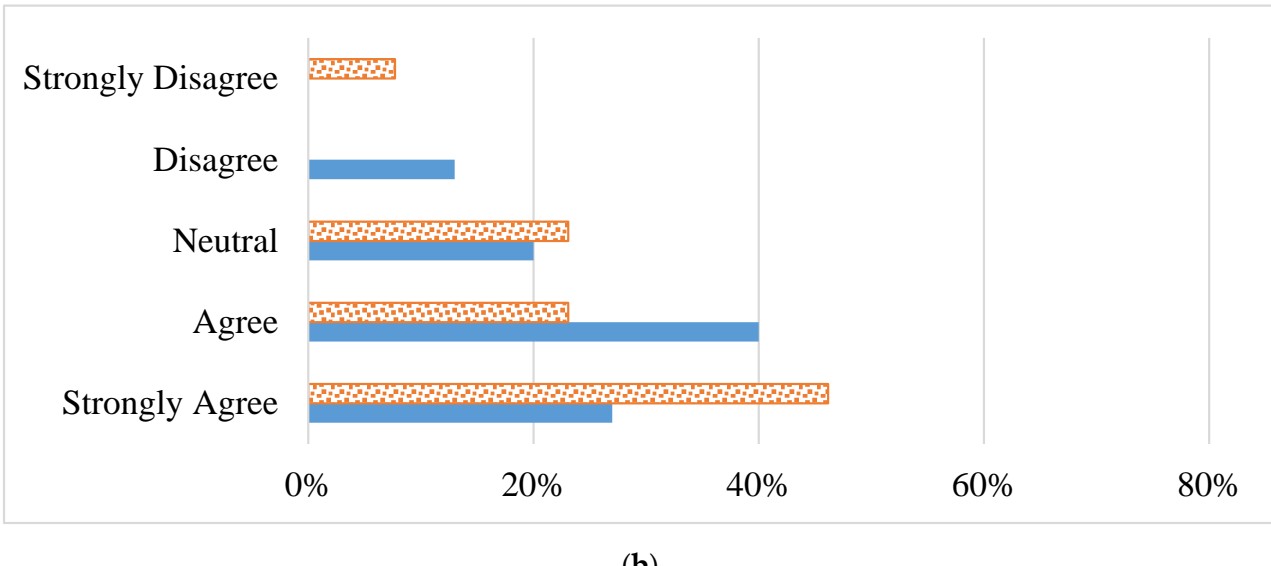

(**b**)

**Figure 4.** Benefits to practitioners and practitioners' organizations: (**a**) mentor or mentor's company receives additional technical resources dedicated to solving a technical problem at a lower cost; (**b**) working with engineering students can provide the mentor's organization with a new way of looking at and solving problems.

The practitioner survey had open-ended questions about the benefits to practitioners and practitioners' organizations. In an open-ended question "*what did you like most about your involvement?*," the frequent responses were: (i) the opportunity to expose senior students to real-world design practice; (ii) engaging with and challenging students; (iii) the ability of the students to provide sound engineering design throughout the process; (iv) gaining a better understanding of the civil and environmental engineering programs at FGCU. Similarly, (i) student work products helped inform actual project development; (ii) obtaining a couple of solutions/alternative designs from students; (iii) hiring graduates were typical responses to the question "*how did you or your organization benefit from your mentorship?*" These findings supported practitioners' benefits pointed out by Akili [15], including the "academic setting" for practitioners to express their views and exchange design concepts, "to help recruit graduates," and "industry driven education".

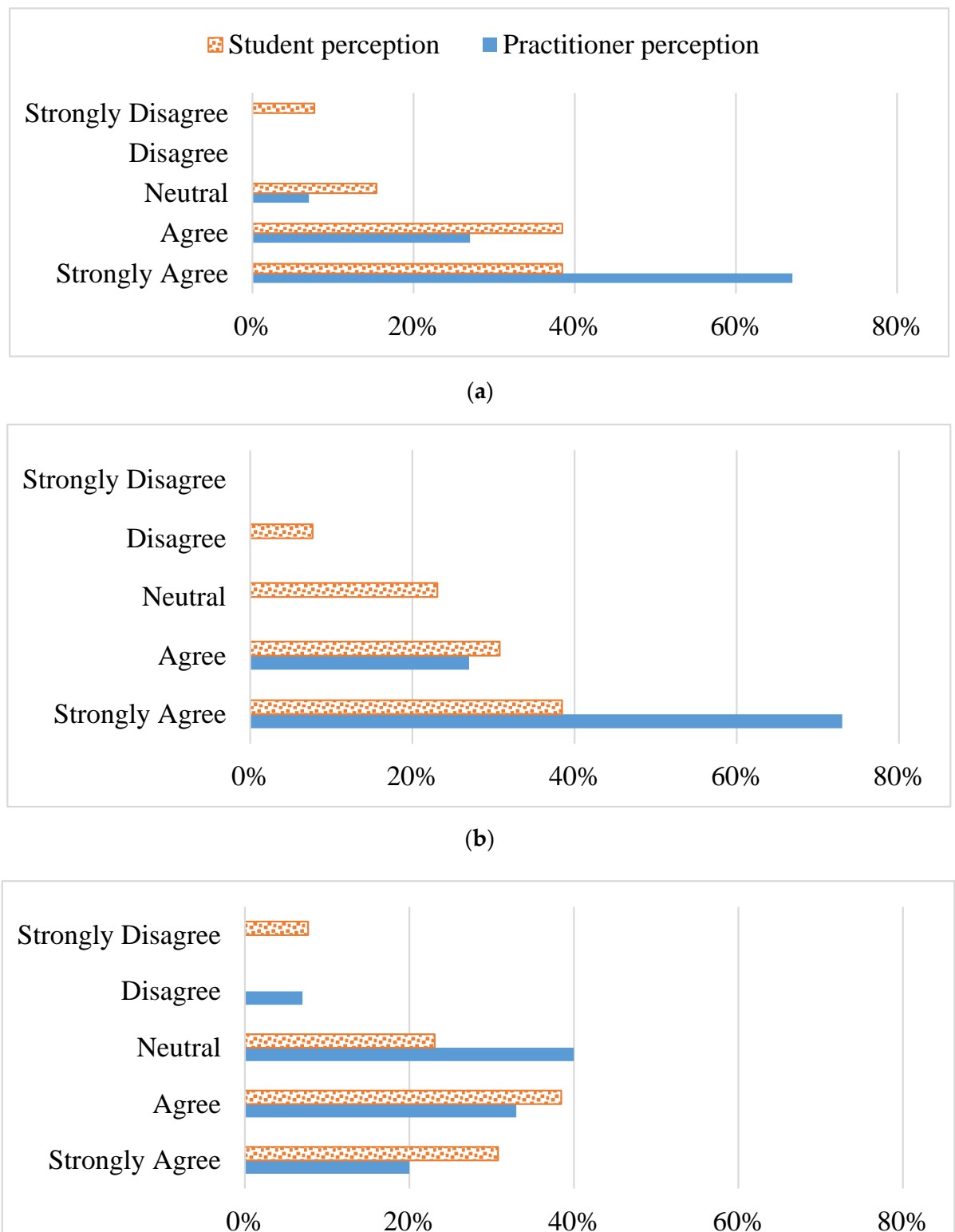

**Figure 5.** Benefits to students due to industry involvement: (**a**) bridging the gap between what is learned in the engineering curriculum and what is expected of graduates when they work in industry; (**b**) exposing students to professional practice; (**c**) students acquire and practice many of the professional skills that are used on a daily basis by design engineers and practitioners.

The mentors, judges, and clients were also asked to provide their overall assessment of the ability of students in senior capstone design. Table 4 summarizes the results. Most of the assessment categories in Table 4 were adopted from Fiegel and Denatale [25]. In general, practitioners are evaluated highly in most assessment categories such as "*conduct research for their projects using library and/or electronic resources*" and "*identify and correctly interpret the relevant codes and standards for their projects*." Categories such as "*evaluate the reasonableness of their design solutions relative to constructability, cost, regulatory environment, etc.*" and "*produce engineering design sketches and/or drawings*" were not rated as high as other categories. These results were in accordance with senior design panel member evaluations discussed in previous work [25]. This feedback undoubtedly helps the department and faculty members identify specific areas to improve student outcomes.

**Table 4.** The ability of students in capstone design.

| The Ability of the Students to: | Poor | Fair | Good | Very Good | Excellent |
|---|---|---|---|---|---|
| Recognize and incorporate the different design constraints of their projects | 0% | 7% | 40% | 40% | 13% |
| Conduct research for their projects using library and/or electronic resources | 0% | 7% | 27% | 33% | 33% |
| Identify and correctly interpret the relevant engineering codes and standards for their projects | 0% | 7% | 20% | 60% | 13% |
| Interpret and assess data provided for their projects (e.g., lab test results, field test results, topography, traffic data, as-built plans, etc.) | 0% | 7% | 33% | 47% | 13% |
| Evaluate the reasonableness of their design solutions relative to constructability, cost, regulatory environment, etc. | 7% | 36% | 29% | 21% | 7% |
| Assess the impact that their design solutions will have on the local environment | 0% | 20% | 27% | 33% | 20% |
| Use computational software and/or spreadsheets to support their design calculations | 0% | 21% | 14% | 36% | 29% |
| Produce engineering design sketches and/or drawings | 8% | 23% | 15% | 38% | 15% |
| Possess communication skills with regard to the presentations | 0% | 0% | 57% | 43% | 0% |

In addition to the official SPoI survey discussed previously, instructors conducted an informal student survey at the end of the course. This survey included the student self-assessment of the course objectives, the contribution of the course to the student's ability to accomplish the program outcomes, and a series of questions about the course materials, assignments, and so on. Figure 6 summarizes the responses to eight questions that are relevant to the purpose of this paper. Students were asked on a 5-point scale: 1 = "strongly disagree"; 2 = "disagree"; 3 = "neutral"; 4 = "agree"; 5 = "strongly disagree." The horizontal axis reports response means.

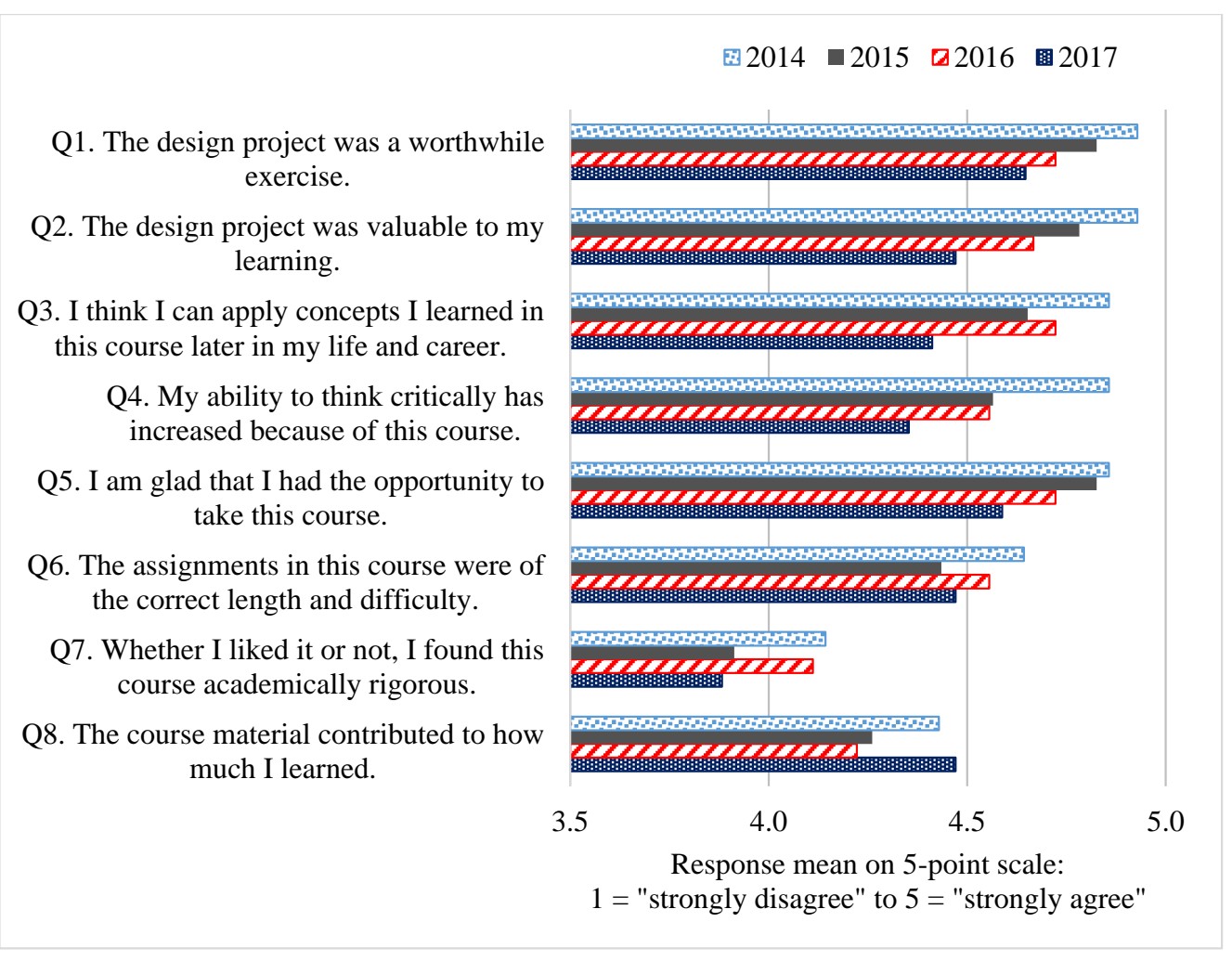

**Figure 6.** Student rating to eight sample questions.

Although this informal survey was not intended to directly assess the contribution of the industry involvement, students tended to highly rate the multiple aspects of the capstone course in four years (2014–2017), especially statements related to design projects (Q1 and Q2 in Figure 6). The consistently high ratings of "*the design project was a worthwhile exercise*" (Q1) and "*the design project was valuable to my learning*" (Q2) demonstrate the valuable involvement and contribution of the practitioners to student learning in these capstone design courses.

**4. Success Stories**

Various capstone projects have been very successful in the past years due to the contribution and participation of local professionals and organizations. The two projects presented here both benefited from the participation of practitioner advisors. In a civil engineering capstone project in the spring of 2015, a group of four students conducted a redesign of the 10-acre Ponce de Leon Park for the City of Punta Gorda, FL. The project included relocating a wildlife center into the park property and relocating/redesigning pavilions, restrooms, parking, playground, and other park amenities. City staff provided the project program, guided site visits, and provided assistance as if the students' team were the city's hired design-engineering consultants. Students presented their designs twice in front of the city's Development Review Committee, concerned citizens, and local media (Figure 7; FGCU, 2016). The student work and final design for this project helped inform the actual project development for the city and this project was under design as of March 2018.

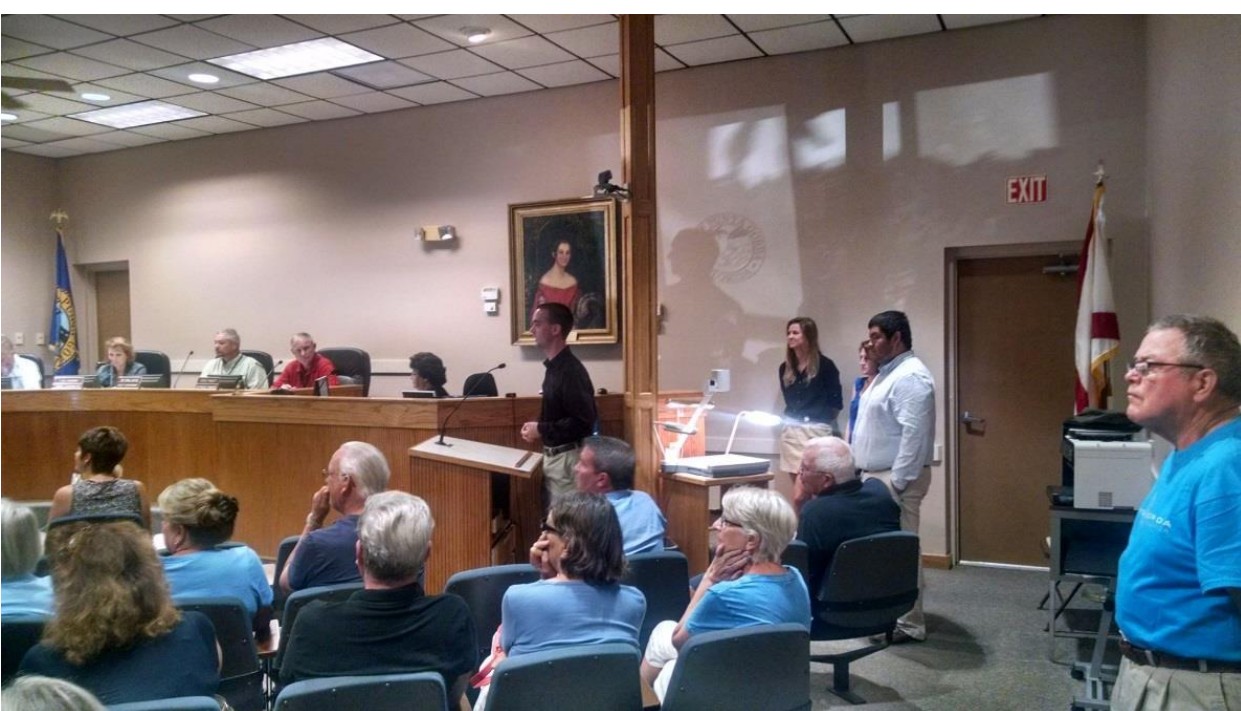

**Figure 7.** Presentations by students to the City's Development Review Committee meeting. (Photo courtesy of the City of Punta Gorda, FL, USA).

Environmental Engineering students routinely present their projects at the Florida Water Environment Association's Student Design Competition. In 2015, the FGCU student group won the competition for their project designing and operating a bioreactor treatment system that effectively treated wastes generated in the biotechnology labs on campus. The students evaluated the impacts of the process using a life cycle approach to show the additional benefits of saving off-site disposal costs and promoting campus sustainability. They went on to present their work at the annual Water Environment Federation annual conference in Chicago, where they won second place overall in the national student design competition, competing against teams comprised of graduate students from universities around the country.

## 5. Discussion

The involvement of industry practitioners in capstone design projects has undoubtedly benefited student learning and to some extent practitioners' organizations. The feedback from both students and practitioners confirmed its contribution. However, the data showed that there was a gap in perception between students and practitioners with regard to the benefits of industry involvement in several aspects. Students tended to rate the benefits to practitioners or practitioners' organizations higher while practitioners tended to rate the benefits to students higher. Closing this gap may help the industry involvement be more fruitful in capstone design projects and courses. That is, when both students and practitioners consider the industry involvement highly beneficial to both mentees and mentors, the mentee–mentor relationship and, therefore, student work products are more likely to succeed.

The results have some implications for faculty teaching these courses. They showed what students did well and pinpointed areas for improvement through the lens of industry practitioners. The perspectives of practitioners could also reflect those of potential employers. Specifically, although industry involvement in capstone design projects has many benefits, it may not substantially help students in acquiring and practicing professional skills that are used on a daily basis by design engineers, as shown in Figure 5c. Although students were eager to apply their technical skills in capstone design projects, proficiency

with professional skills could present a major obstacle [26]. This lack of professional skills was indicated by the lower practitioner rating of student ability in related categories such as "*evaluate the reasonableness of their design solutions relative to constructability, cost, regulatory environment, etc.*" and "*produce engineering design sketches and/or drawings*" (Table 4). This implied that internship programs play a critical role in improving those categories of student ability. Industrial experience such as internships can shape how engineering seniors frame design as a whole system, including uncertainties, challenges, design tools and analyses, economics, regulations, and so on [27]. As instructors, the authors observed that student teams where members participated in relevant internships tended to perform better in their design products, including the above ability categories. In the future, further studies should be conducted to examine the impact of internships on student success in capstone programs.

Finally, the level of success has varied from project to project. As instructors, the authors have observed that there are common drivers that can bring the success of the senior capstone design projects out of the classroom boundary such as the two success stories presented in the previous section. They are: (i) projects are authentic, in their conceptual and early development stage and from the mentors' organizations; (ii) student teams are highly committed to the projects throughout the semester; (iii) mentors are available and responsive to students, have internal support, and value student work; (iv) instructors constantly communicate with mentors and student teams, facilitate the student–mentor relationship, and monitor the design and development progress of each student team. As each project has its unique type, constraints, and scope of design or experiments, team office hours or interactive questions and answers (Q&A) sessions that are periodically scheduled throughout the semester between each team and instructor are very helpful. Instructors can use these team office hours and Q&A sessions to better control project progress, address student needs and resolve team conflict and misunderstandings between mentor and student team that may arise. In addition to peer evaluation, team office hours and Q&A sessions are also a means for instructors to assess how and to what extent each team member contributes to teamwork products. This is because it is always a challenge for instructors to assess the performance of each individual student in a project-based learning environment such as senior capstone design. Lastly, the timing of industry participation and the arrangement of student teams can also be challenging [28].

## 6. Conclusions

At Florida Gulf Coast University, industry contribution to the senior design capstone courses in the Department of Environmental and Civil Engineering has played a crucial role in the students' design education for many years [29]. Local professional engineers and practitioners have served as project mentors and judges of the final design poster presentations. Students benefit from working on real-world problems, with better preparation for a career in engineering, and the possibility of being employed by their mentors after graduation. Two-thirds of practitioners agreed or strongly agreed that working with engineering students can provide their organization with a new way of looking at and solving problems. Through their involvement, practitioners would provide valuable feedback that helps department and faculty members identify and emphasize specific knowledge areas taught in different courses to improve the performance of students in their senior capstone design and their engineering career.

Through the collection of feedback from both students and practitioners in addition to faculty reflection, this paper contributed to the extant literature by showing that there was a considerable gap in perceptions between engineering seniors and practitioners with regard to the benefits of industry involvement in capstone design courses. The authors also suggested the success drivers for capstone design projects from the collective reflection of instructors teaching this course. Closing the gap in perceptions between students and

practitioners and fostering the success drivers may bring the results of capstone design projects out of the classroom boundary.

**Author Contributions:** Conceptualization, R.O., S.K. and A.B.; Methodology, S.K., A.B. and J.-Y.K.; software, A.B. and K.-D.K.; validation, R.O., S.K., J.-Y.K., K.-D.K. and A.B.; formal analysis, A.B. and J.-Y.K.; investigation, R.O., S.K., J.-Y.K., K.-D.K. and A.B.; resources, R.O. and S.K.; data curation, A.B. and J.-Y.K.; writing—original draft preparation, R.O., S.K., J.-Y.K., K.-D.K. and A.B.; writing—review and editing, R.O., S.K., J.-Y.K., K.-D.K. and A.B.; visualization, A.B., J.-Y.K. and K.-D.K.; supervision, R.O., S.K. and A.B.; project administration, K.-D.K. and A.B.; All authors have read and agreed to the published version of the manuscript.

**Funding:** This research received no external funding.

**Conflicts of Interest:** The authors declare no conflict of interest.

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
