# Peer review of "Fostering Project-Based Learning through Industry Engagement in Capstone Design Projects"

_education, doi:10.3390/educsci13040361_

Round 1
Reviewer 1 Report
GENERAL:
- This is a great topic and it is great to cover this aspect in civil engineering in terms of perspectives. Because of the academic source of projects -offered in a university setting- I would not necessarily agree that Capstone projects propose to replicate real-world experience in it their entirety. There are memos, papers, writing assignments, activities, etc that may not be seen in the industry and yet they are necessarily part of the Capstone.
- Nice overview of project types - These tables provide a context for the rest of the paper.
- You mention "Feedback from both students and practitioners were collected to gain some insights 152 into the industry involvement in these capstone courses" Did you have theories? Was there a problem to solve? An observation to test? Why [exactly] were you asking these questions? It appears to be related to your remark "employers have observed a big gap between expectation and reality"
- The role of mentors or mentors-practitioners was mentioned a lot, but I could not find an actual definition of the mentor's role
- It does not look like you asked about the industry client's involvement in Capstone "in order to get a project done". That is common among the top objectives -along with others.
- This work is valuable in helping to align students and practitioners alike to help calibrate expectations on both sides and prepare each constituent for the best outcome.
- The conclusion "The results have some implications for faculty teaching these courses. They showed what students did well and pinpointed areas for improvement through the lens of industry practitioners" makes this meaningful work for Faculty Capstone coordinators in the specific areas outlined. I would recommend placing this outcome clearly in the abstract.
FORMATTING:
- Need a line feed between lines 71 & 72, 114 & 115, 288 & 289
- Try not to split a table across pages when it will fit on one page.
- Move the heading "Feedback of Students and Practitioners" to be with the body of the section
REFERENCES:
- You have selected some good relevant resources, but not many that are recent. There is a percentage from 2017 and one from 2018, but many are from the early 2000s and some earlier in areas that have been re-researched.
- A suggestion would be to search further, especially in conference proceedings, like ASEE and Journal of Education as well as special editions for Capstone topics, like the International Journal of Education.
Author Response
GENERAL:
- This is a great topic and it is great to cover this aspect in civil engineering in terms of perspectives. Because of the academic source of projects -offered in a university setting- I would not necessarily agree that Capstone projects propose to replicate real-world experience in it their entirety. There are memos, papers, writing assignments, activities, etc that may not be seen in the industry and yet they are necessarily part of the Capstone.
Comment: We appreciate the kind words about the topic and our paper.
Added to provide clarity about overlap between capstone and industry: The role of the practitioners was to mentor the students through the design process including project initiation, scope of work, project planning, preliminary, intermediate and final design work as well as final presentations.
- Nice overview of project types - These tables provide a context for the rest of the paper.
Response: We appreciate your kind comment.
- You mention "Feedback from both students and practitioners were collected to gain some insights 152 into the industry involvement in these capstone courses" Did you have theories? Was there a problem to solve? An observation to test? Why [exactly] were you asking these questions? It appears to be related to your remark "employers have observed a big gap between expectation and reality"
Comment: We were indeed asking these questions because our primary mission is to prepare engineers for successful careers in industry.
The following section was edited to provide clarity: In engineering education, although there is high degree of agreement on the importance of professional skills (interpersonal skills, teamwork, communication, and problem-solving skills), employers have observed a big gap between expectation and reality (Chan et al. 2017; Craps et al. 2017). This was one of the primary reasons for completing this study as our main goal as educators is to prepare engineer students for successful careers in industry. With regard to engineering practice, literature has shown significant differences in rigor of design and professionalism between school and work (Paretti et al. 2017). Therefore, an affiliation of active engineers and professionals from industry would strengthen the professional elements of engineering education (EdstrÓ§m, 2018).
- The role of mentors or mentors-practitioners was mentioned a lot, but I could not find an actual definition of the mentor's role
The following was added for clarity: The role of the practitioners was to mentor the students through the design process including project initiation, scope of work, project planning, preliminary, intermediate and final design work as well as final presentations.
- It does not look like you asked about the industry client's involvement in Capstone "in order to get a project done". That is common among the top objectives -along with others.
Comment: We did not ask this question because in our experience clients do not interact with students to get a project done. If they need to get a project done they keep it in house and quickly finish it. The main reasons as we explain in the article are that clients want to recruit students and give back to the engineering community.
- This work is valuable in helping to align students and practitioners alike to help calibrate expectations on both sides and prepare each constituent for the best outcome.
Response: We appreciate the kind comment. We also feel like aligning expectations is crucial and hope that this article will help other programs do this.
- The conclusion "The results have some implications for faculty teaching these courses. They showed what students did well and pinpointed areas for improvement through the lens of industry practitioners" makes this meaningful work for Faculty Capstone coordinators in the specific areas outlined. I would recommend placing this outcome clearly in the abstract.
The following was added to the abstract: The results of this study have substantial implications for faculty teaching these courses. They showed what students did well and pinpointed areas for improvement through the lens of industry practitioners.
FORMATTING:
- Need a line feed between lines 71 & 72, 114 & 115, 288 & 289
Response: All of the line feeds have been added at the indicated locations
- Try not to split a table across pages when it will fit on one page.
Comment: All tables were moved so that they all appear on one page
- Move the heading "Feedback of Students and Practitioners" to be with the body of the section
Comment: The heading was moved to be with the body of the section
REFERENCES:
- You have selected some good relevant resources, but not many that are recent. There is a percentage from 2017 and one from 2018, but many are from the early 2000s and some earlier in areas that have been re-researched.
- A suggestion would be to search further, especially in conference proceedings, like ASEE and Journal of Education as well as special editions for Capstone topics, like the International Journal of Education.
Comment: The following more up to date references were added throughout the text where appropriate.
Guanes, G., Wang, L., Delaine, D.A. and Dringenberg, E., 2022. “Empathic approaches in engineering capstone design projects: student beliefs and reported behaviour.” European Journal of Engineering Education, 47(3), pp.429-445.
Meah, K., Hake, D. and Wilkerson, S.D.,. “A multidisciplinary capstone design project to satisfy ABET student outcomes.” Education Research International, 2020, pp.1-17.
Howe, S., and Goldberg, J. “Engineering Capstone Design Education: Current Practices, Emerging Trends, and Successful Strategies.” In: Schaefer, D., Coates, G., Eckert, C. (eds) Design Education Today. Springer, Cham. 2019. https://doi.org/10.1007/978-3-030-17134-6_6
Ozkan, D.S., Murzi, H.M., Salado, A., and Gerwitz, C. “Reality Gaps in Industrial Engineering Senior Design or Capstone Projects,” Proceedings of the American Society for Engineering Education Conference, New Orleans, LA, June 2018.
Goldberg, J., Cariapa, V., Corliss, G. and Kaiser, K., “Benefits of industry involvement in multidisciplinary capstone design courses,” International Journal of Engineering Education, 30(1), pp. 6-13, 2014.
Jaeger, B.K., and Smyser, B.M., “Student-Generated Metrics as a Predictor of Success in Capstone Design,” Proceedings of the American Society for Engineering Education Conference, Indianapolis, IN, June 2014.
Reviewer 2 Report
Overall comments:
A The authors present helpful survey results and share gained knowledge from faculty experience from implementing practitioner-mentored capstone projects over a several years. While useful, the paper seems a bit lengthy considering the results presented. Additionally, the article reads in various places as an advertisement for a particular university which is concerning. With revision and reasonable responses to the inquiries below I believe this paper is a good addition to the existing literature.
B Need to make it clearer what the studied period was throughout the paper. In some cases, it seems like data was collected from 2014-2017. In others you state that a new survey metric was put in place starting in 2017. The timeline in not clear to the reader.
C The insights provided at the end of the discussion section as to best practices for successful implementation of practitioner-led capstone projects are very useful. I would’ve like to see more focus on this aspect, with data to measure these effects.
Detailed comments:
1. Lines 50-51: it would be helpful to elaborate on what these gaps were found to be
2. Line 92-93: have you included and clearly described the findings in these ‘few’ studies?
3. Lines 141-143: I don’t see the purpose of this sentence. It reads like an ad for FGCU and doesn’t flow with surrounding sentences.
4. Line 148: The ABET student outcomes in Table 3 appear to be refencing an outdated ABET standard. The new standard only includes 7 student outcomes. Is there a reason for using the old method? Either way the outcomes should be clearly linked to an ABET citation with a bit more explanation.
5. Line 153: Need to clarify: was the survey sent out to all participants over the last x years, or a subset? If a subset, then how were they selected and how is sampling bias accounted for?
6. Line 163: 40+7 = 47%, not 60% ??
7. Line 164: 57% of those who mentored or of all who were involved?
8. Line 165: Did any practitioners mentor more than 1 project at a time?
9. Line 248: The first example here is clear and useful. Was there practitioner participation in the 2nd example of the bioreactor? If not, is it relevant here? Congratulations to those students, but I don’t see the link currently. It seems like it would be equally if not more helpful to also include lessons learned from projects that proved difficult or were otherwise ‘unsuccessful’ in some way.
10. Line 260: Any comment on the clearly decrease in values over the 4 years in Q1-Q5?
11. Line 297: this was additionally indicated?
12. Line 300: this inference is not however proved by the data in this study, correct? You go on to use instructor experience as evidence, which is fine, but it should be made clear that other studies are needed to prove this.
13. Line 304: typo
Author Response
A The authors present helpful survey results and share gained knowledge from faculty experience from implementing practitioner-mentored capstone projects over a several years. While useful, the paper seems a bit lengthy considering the results presented. Additionally, the article reads in various places as an advertisement for a particular university which is concerning. With revision and reasonable responses to the inquiries below I believe this paper is a good addition to the existing literature.
Response: We appreciate the kind comments of the reviewer. We have addressed all of the comments below and removed sentences that read like an advertisement for FGCU
The following sentence was removed: FGCU a teaching focused university located in a mid-size city, 80% of the classes are taught by full time faculty.
Lines 141-143: I don’t see the purpose of this sentence. It reads like an ad for FGCU and doesn’t flow with surrounding sentences.
Changed to read: The accreditation of engineering programs necessitates design experience; and recognizes the value of industry partners in the capstone design courses in preparing students for careers in engineering fields.
B Need to make it clearer what the studied period was throughout the paper. In some cases, it seems like data was collected from 2014-2017. In others you state that a new survey metric was put in place starting in 2017. The timeline in not clear to the reader.
The following sentences were modified to provide clarity:
A survey questionnaire was designed and sent to all 26 practitioners who were mentors, judges, and/or clients between 2014 and 20187 and encompassed 4 capstone cohorts.
The average total course enrollment in the five spring semesters (2014 - 2017) was 56 students.
For the last year of the study period, questions regarding the benefits of industry involvement were also added in the official end-of-semester student survey administered by the university, called student perception of instruction (SPoI).
Although this informal survey was not intended to directly assess the contribution of the industry involvement, students tended to highly rate the multiple aspects of the capstone course in 4 years (2014-2017), especially statements related to design projects (Q1 and Q2 in Figure 6).
C The insights provided at the end of the discussion section as to best practices for successful implementation of practitioner-led capstone projects are very useful. I would’ve like to see more focus on this aspect, with data to measure these effects.
Comment: We are currently working on a follow up study to examine exactly these effects. We hope to be submitting a manuscript with these within the net two years once we have collected substantial data.
Detailed comments:
- Lines 50-51: it would be helpful to elaborate on what these gaps were found to be
Changed to read: With regard to engineering practice, literature has shown significant differences in rigor of design and professionalism between school and work
- Line 92-93: have you included and clearly described the findings in these ‘few’ studies?
Changed to Read: Limited studies have addressed perspectives of both students and practitioners with respect to the benefits of industry engagement and possible differences between the two groups. The studies that do exist have addressed student perception and success and industry involvement (Goldberg et al. 2014; Ozkan et al. 2018; Jaeger and Smyser, 2014). This paper aims at filling this gap in the literature by identifying the contributions of industry participants to student experiential learning from the perspectives of both students at FGCU and practitioners.
- Lines 141-143: I don’t see the purpose of this sentence. It reads like an ad for FGCU and doesn’t flow with surrounding sentences.
Changed to read: The accreditation of engineering programs necessitates design experience; and recognizes the value of industry partners in the capstone design courses in preparing students for careers in engineering fields.
- Line 148: The ABET student outcomes in Table 3 appear to be refencing an outdated ABET standard. The new standard only includes 7 student outcomes. Is there a reason for using the old method? Either way the outcomes should be clearly linked to an ABET citation with a bit more explanation.
Comment: You are correct that we are referencing the old a-k outcomes. The reason for this is that the data for this study was collected under the old a-k outcomes. Due to Covid related delays it has just taken us some time to get this paper put together and sent out for review. The sentences below have been added to provide clarity and an ABET reference has been added.
Changed to read: Table 3 presents the course learning objectives and the corresponding ABET student outcomes. The ABET outcomes in the paper are the old a-k outcomes as this study was conducted before the change to the new 1-7 outcomes (ABET, 2012). These course objectives were assessed by the project deliverables submitted and/or presented as individual and teamwork efforts mentioned above.
- Line 153: Need to clarify: was the survey sent out to all participants over the last x years, or a subset? If a subset, then how were they selected and how is sampling bias accounted for?
Changed to read: A survey questionnaire was designed and sent to all 26 practitioners who were mentors, judges, and/or clients between 2014 and 2017.
- Line 163: 40+7 = 47%, not 60%??
Changed to Read: Most of the practitioners were involved as mentors (7%) or both mentor and judge (40%) (Figure 1)
- Line 164: 57% of those who mentored or of all who were involved?
Changed to read: In addition, 57% of all of the practitioners involved mentored 3 projects or more
- Line 165: Did any practitioners mentor more than 1 project at a time?
Changed to read: This indicated that they were involved in these capstone design courses for several years and some practitioners mentored more than one project at the same time.
- Line 248: The first example here is clear and useful. Was there practitioner participation in the 2ndexample of the bioreactor? If not, is it relevant here? Congratulations to those students, but I don’t see the link currently. It seems like it would be equally if not more helpful to also include lessons learned from projects that proved difficult or were otherwise ‘unsuccessful’ in some way.
Comment: There was practitioner advisor participation in this project.
Changed to read: Various capstone projects have been very successful in the past years due to the contribution and participation of local professionals and organizations. The two projects presented here both benefited from the participation of practitioner advisors.
Comment: While we agree that including some unsuccessful projects would be beneficial here we did not collect much data nor receive significant comments about such projects. This is something we plan to include in future studies.
- Line 260: Any comment on the clearly decrease in values over the 4 years in Q1-Q5?
Comment: While there is a decrease in the four years for questions Q1-Q5 that decrease is minimal and cannot be explained with the collected data. We have observed an increase in student negativity and cynicism as students are asked to continually pay more for their education. Students are increasingly working more so they can pay for college, have less time for doing assignments, and generally have a less positive outlook for the future. Student satisfaction at many universities has been declining for multiple reasons. This decline merits study and we plan to incorporate this in our next paper that we are currently working on.
- Line 297: this was additionally indicated?
Changed to read: This lack in professional skills was indicated from the lower practitioner rating of student ability in related categories such as “evaluate the reasonableness of their design solutions relative to constructability, cost, regulatory environment, etc.” and “produce engineering design sketches and/or drawings”
- Line 300: this inference is not however proved by the data in this study, correct? You go on to use instructor experience as evidence, which is fine, but it should be made clear that other studies are needed to prove this.
Changed to read: As instructors, the authors observed that student teams where members did some relevant internships tended to perform better in their design products, including the above ability categories. In the future further studies should be conducted to examine the impact of internships on student success in capstone programs.
- Line 304: typo
Typo Fixed to read: As instructors, the authors observed that student teams where members did some relevant internships tended to perform better in their design products, including the above ability categories.